# Transcriptomic Analysis Reveals Dysregulation of the Mycobiome and Archaeome and Distinct Oncogenic Characteristics according to Subtype and Gender in Papillary Thyroid Carcinoma

**DOI:** 10.3390/ijms24043148

**Published:** 2023-02-05

**Authors:** Daniel John, Rishabh Yalamarty, Armon Barakchi, Tianyi Chen, Jaideep Chakladar, Wei Tse Li, Weg M. Ongkeko

**Affiliations:** 1Division of Otolaryngology-Head and Neck Surgery, Department of Surgery, UC San Diego School of Medicine, San Diego, CA 92093, USA; 2Research Service, VA San Diego Healthcare System, San Diego, CA 92161, USA

**Keywords:** Papillary Thyroid Carcinoma (PTC), microbiome, mycobiome, fungi, archaeome, BRAF, RET/PTC

## Abstract

Papillary Thyroid Carcinoma (PTC) is characterized by unique tumor morphology, treatment response, and patient outcomes according to subtype and gender. While previous studies have implicated the intratumor bacterial microbiome in the incidence and progression of PTC, few studies have investigated the potential role of fungal and archaeal species in oncogenesis. In this study, we aimed to characterize the intratumor mycobiome and archaeometry in PTC with respect to its three primary subtypes: Classical (CPTC), Follicular Variant (FVPTC), and Tall Cell (TCPTC), and also with respect to gender. RNA-sequencing data were downloaded from The Cancer Genome Atlas (TCGA), including 453 primary tumor tissue samples and 54 adjacent solid tissue normal samples. The PathoScope 2.0 framework was used to extract fungal and archaeal microbial read counts from raw RNA-sequencing data. Overall, we found that the intratumor mycobiome and archaeometry share significant similarities in CPTC, FVPTC, and TCPTC, although most dysregulated species in CPTC are underabundant compared to normal. Furthermore, differences between the mycobiome and archaeometry were more significant between males and females, with a disproportionate number of fungal species overabundant in female tumor samples. Additionally, the expression of oncogenic PTC pathways was distinct across CPTC, FVPTC, and TCPTC, indicating that these microbes may uniquely contribute to PTC pathogenesis in each subtype. Furthermore, differences in the expression of these pathways were observed between males and females. Finally, we found a specific panel of fungi to be dysregulated in BRAF V600E-positive tumors. This study demonstrates the potential importance of microbial species to PTC incidence and oncogenesis.

## 1. Introduction

Thyroid cancer is the fastest-growing cancer in the world [1,2] and the fifth most common cancer in women [3], accounting for more than 586,000 new cases in 2020 [2]. Papillary Thyroid Carcinoma (PTC) accounts for more than 88% of total thyroid cancer cases and consists of three primary variants, each of which is characterized by a unique treatment course and prognosis [4,5]. Classical Papillary Thyroid Carcinoma (CPTC) is the most common subtype, consisting of more than 67% of total PTC diagnoses in the US from 2000–2017 [6]. While the standard definition has varied frequently, Follicular Variant (FVPTC) prognostically lies between CPTC and Follicular Papillary Thyroid Carcinoma (FPTC) and is associated with a high mortality rate and increased metastasis compared to CPTC [7]. Although much less frequent in incidence, the Tall Cell (TCPTC) variant is considered one of the most aggressive forms of PTC, exhibiting significantly poorer prognosis and 5-year survival rates compared to classical forms of PTC [8,9]. 

While only a few risk factors for PTC are understood, gender is a defining feature of PTC occurrence and development [10]. Female patients are three times more likely to be diagnosed with PTC than males [11,12]. Despite the significantly higher incidence rates of PTC in females, the prognostic significance of gender in PTC does not appear to follow the same trend. In a retrospective cohort study of 3572 PTC patients, it was found that overall survival outcomes were comparable between men and women. An increased Hazards Ratio (HR) was observed for men diagnosed before 55 years of age, whereas the HR was similar for men and women diagnosed between 55–69 years of age [12]. Similarly, a comprehensive study of 43,712 PTC patients in the US using the National Cancer Institute’s (NCI) Surveillance, Epidemiology, and End Results (SEER) cancer registry found significantly elevated mortality among male PTC patients after adjusting for confounding variables [13]. It is worth noting that these gender differences are primarily prevalent in CPTC, whereas patients with more lethal variants of PTC experience similar incidence rates and survival characteristics across gender comparisons [14]. Several potential mechanisms, such as differences in sex hormones and reproductive factors [11], have been proposed as an explanation for this disparity, but studies have not conclusively proven a mechanism for these differences. Thus, while PTC subtype and gender disparities in PTC incidence and outcomes are clearly evident, more research is needed to elucidate how such differences contribute to PTC oncogenesis.

Studies of the human microbiome have revealed its important implications on the human body and the pathogenesis of a variety of diseases [15,16]. While early studies of the microbiome focused primarily on the interactions of gut flora with innate phenotypes [17,18], novel studies have characterized the microbiome as an important driver of various diseases, including inflammatory bowel disease (IBD) [19,20], arthritis [21,22], Alzheimer’s [23,24], diabetes [25,26], cardiovascular disease [27,28] and cancer [29]. The intratumor microbiome in cancer has emerged as an important player in mediating effective immune response, treatment efficacy, and cell survival through the production of metabolites and downstream interactions within the tumor microenvironment [30,31,32]. Previously, we have implicated the intratumor microbiome in head and neck [33], pancreas [34], prostate [35], lung [36], and bladder [37] cancers, particularly in initiating oncogenesis through immune response, mutation events, methylation, microRNAs (miRNA) and more.

In addition to these studies, we also characterized the intratumor bacterial microbiome in PTC, demonstrating its unique implications for PTC prognosis through immunological pathways, mutation events, and gene methylation. Importantly, we found significantly distinct microbial and potential mechanistic landscapes across PTC subtype and gender comparisons, indicating that these factors likely serve as an important modulator of the intratumor microbiome in PTC. Additionally, we also found unique oncogenic signature pathways associated with each PTC subtype, further indicating the importance of studying the transcriptomic landscape according to these clinical characteristics [38]. 

Given our previous findings of the bacterial microbiome in PTC, we aimed to further investigate intratumor fungal and archaeal microbes according to PTC subtype and gender comparisons. In this study, we characterized the similarities and differences in the PTC mycobiome and archaeometry according to PTC subtype and gender comparisons. We first identified significantly dysregulated microbes according to these cohorts, assessed the relevance of these microbes with patient clinical variables, investigated their correlations with known driver gene signature pathways of PTC oncogenesis, such as BRAF and RET/PTC [39,40], and finally studied the association of these intratumor microbes with the presence of the BRAF V600E mutation (Figure 1).

## 2. Results

### 2.1. Data Acquisition and Extraction Identification of Microbial Reads 

In order to evaluate intratumor fungal and archaeal microbial species pertinent to PTC, we downloaded raw RNA-sequencing data for 453 PTC patients and 54 adjacent normals from The Cancer Genome Atlas (TCGA). Fungal and Archaeal microbial read counts were extracted from RNA-sequencing data using the PathoScope 2.0 framework. Clinical data for patients in the TCGA-THCA project were downloaded from the Broad Institute GDAC Firehose Database (https://gdac.broadinstitute.org/), accessed on 15 January 2022. 

### 2.2. Removal of Potential Contaminants

Following the successful extraction of microbial read counts, we identified potential contaminants using three methods of contamination correction. Selected plots generated in this analysis are shown in Figure 2. A full list of fungal and archaeal contaminants identified in our study is included in Appendix A. First, we identified six potential archaea contaminants using correction by sequencing date (Figure 2D). We did not identify any potential fungal contaminants using correction by sequencing date (Figure 2A). Next, we evaluated potential fungal and archaeal contaminants likely introduced by sequencing plates. Our analysis did not, however, identify any fungal or archaeal contaminants using correction by sequencing plate (Figure 2B,E). Finally, we identified 1492 fungal and 286 potential archaeal contaminants using correction by total microbe abundance (Figure 2C,F). In total, we identified 1492 and 292 potential fungal and archaeal contaminants, respectively. These microbes were removed from the original 9829 fungal and 483 archaeal species extracted for downstream analyses.

### 2.3. Differentially Abundant Fungal and Archaeal Species across PTC Subtype and Gender Comparisons

In order to characterize the role of the intratumor mycobiome and archaeometry in PTC, we first identified and compared significantly dysregulated fungal and archaeal species across PTC subtype and gender comparisons. Significantly differentially abundant microbes were defined by a *p*-value of <0.05 and a log-fold change (FC) > 1. The Bonferroni method was used to correct *p*-values for multiple hypothesis testing. We first identified 109 significantly dysregulated microbes between tumor and normal tissue (Appendix A). Of these significantly dysregulated species, 14 fungi were overabundant in tumor tissue, while 94 fungi and one archaeal species were overabundant in normal tissue. Fungal species overabundant in PTC tumor tissue included *Metarhizium acridum CQMa 102*, *Saccharomyces cerevisiae YJM1338*, and *Phaffia rhodozyma*, which we similarly found to be abundant in HNSCC tumor tissue [33]. Additionally, the archaeal species *Anomalluma dodsoniana* was overabundant in PTC tumor tissue compared to adjacent normal. We found significantly more species overabundant in normal tissue, however, including *Candida albicans*, *Microallomyces dendroideus*, and the archaeal species *Anomalluma dodsoniana*. 

Additionally, we identified similarly significantly dysregulated fungal and archaeal microbes in comparisons of CPTC vs normal samples, FCPTC vs normal samples, and TCPTC vs normal samples with respect to the direction of dysregulation. Select microbes and corresponding overlaps across these subtypes are visualized in Figure 3, and the full list of significantly dysregulated microbes is included in Appendix A. In total, 63 fungal species were overabundant in CPTC, FVPTC, and TCPTC, including *Botrytis cinerea*, *Pichia cephalocereana,* and *Trematosphaeria pertusa* (Figure 3A,C, Appendix A). The majority of dysregulated fungal and archaeal species were overabundant in comparisons of PTC subtypes with normal samples. Specifically, 24 fungal species were overabundant in TCPTC and FVPTC patients, 3 fungal species were overabundant in TCPTC and CPTC patients, 12 fungal species were overabundant in FVPTC and CPTC patients, 33 fungal species and one archaeal species were overabundant in TCPTC only, 28 fungal and 2 archaeal species were overabundant in FVPTC only, and only one fungal species was overabundant in CPTC only (Figure 3A). Only CPTC and TCPTC subtypes shared similarly underabundant species, which included *Rhizopus arrhizus* and *uncultured Uromyces*. CPTC exhibited the most underabundant species, with 16 underabundant fungi compared to normal samples, while FVPTC samples were significantly underabundant in 6 fungal species, and TCPTC displayed correlations to zero significantly underabundant microbes compared to normal samples. Interestingly, significantly dysregulated archaeal species were distinct in each PTC subtype: *uncultured euryarchaeote Alv-FOS5* was found to be overabundant in TCPTC tumor tissue compared to normal samples (Figure 3C); *uncultured marine archaeon* and *unculture Pyrobaculum* sp. were overabundant in FVPTC tumor tissue compared to normal samples (Figure 3C); while *Halovivax ruber XH-70* and *Methanosarcina* sp. *WH1* were both overabundant in normal samples when compared to

CPTC tumor tissue (Figure 3C). Overall, a large number of microbes were similarly significantly overabundant in all three PTC types, with FVPTC exhibiting the most significant dysregulated microbes, followed by TCPTC and CPTC. While certain commonalities in microbial abundance appear to be present across these PTC subtypes, these findings suggest a unique microbial landscape within the tumor microenvironment, dysregulated by CPTC, FVPTC, and TCPTC variants. 

Given the known importance of gender in PTC incidence and outcomes, we additionally identified similarly significantly dysregulated microbes in male and female PTC tumor samples compared to adjacent gender-controlled tissue normal (Figure 3B, Appendix A). Significant differences were observed in gender comparisons, with a total of 88 significantly dysregulated fungal microbes in females only, compared to only 11 significantly dysregulated fungal and archaeal microbes in males only. A total of 10 fungal species were overabundant in male and female PTC tumor tissues compared to solid tissue normal, including *Coemansia reversa* (Figure 3D), *Pneumocystis carinii,* and *Inosperma maculatum.* A total of 80 fungal species were overabundant in female tumor tissue only, including *Pseudosperma obsoletum*, *Yamdazyma, triangularis*, and *Candida albicans*, while 8 fungal species were underabundant in female tumor tissue when compared to female adjacent normal tissue samples, which included *Thremochaetoides thermophila DSM 1495* (Figure 3D), *Aspergilus fumigatus Af293*, and *Saccharomyces cerevisiae YJM1418* (Figure 3B). We identified four fungal species that were overabundant in male PTC tumor tissue, while five fungal species were underabundant in male PTC tumor tissue, compared to male adjacent normal samples. Interestingly, the only significantly dysregulated archaeal species identified were underabundant in male PTC tumor tissue, which included *Halovivax ruber XH-70* (Figure 3D) and *Natrialba magadii ATCC 43099* (Figure 3B). Thus, our findings indicate that significant dysregulation of intratumor fungal and archaeal species is present according to gender. Interestingly, these microbes were disproportionately dysregulated in female tumor tissue, indicating that these species may play a potential role in increasing the risk for PTC occurrence, which is substantially higher for females. 

### 2.4. Correlation of Significantly Dysregulated Fungal and Archaeal Species to Clinical Variables

Next, we correlated the abundance of significantly dysregulated microbes to clinical variables pertinent to PTC prognosis and outcomes. Specifically, we analyzed the following clinical variables: follow-up vital status, perineural invasion, pathologic stage, and cancer T, N, and M staging. We performed clinical variable analysis using the Kruskal–Wallis test (*p*-value < 0.05) on all patients with PTC (n = 453). 

In our analysis, we found 26 total microbes to be significantly correlated with patient clinical variables (Figure 4). No archaeal species were significantly associated with clinical variables in our analysis. A full list of microbial species significantly correlated with pertinent clinical variables is included in Appendix A. In patients with follow-up vital status, *Brevicellium exile* was overabundant in patients who were deceased compared to those who were still alive. We found that *Piptocephalis corymbifera* and *Zoophthora occcidentalis*, however, were overabundant in PTC patients with neoplasms compared to those with no neoplasms. Interestingly, all microbe correlations with Patholgic staging (including M and N staging) were associated with a higher pathologic stage (Figure 4, Appendix A). We found that *Chaetomium globosum CBS 148.51* abundance was correlated with increasing pathologic stage. In total, 18 fungal species were correlated with a higher pathologic M stage, including *Candida Albicans*, *Eremascus albus*, and *Thanatephorus cucmeris* (Figure 4, Appendix A). Finally, *Wickerhamiella pararugosa*, *uncultured Cryptomycota*, and *Spiromyces aspiralis* were associated with a higher pathologic N stage. The correlation of studied species with a pathologic T stage yielded no significant results. Due to the lack of significantly dysregulated archaeal microbes in differential abundance analysis and correlation with PTC clinical variables, we did not further analyze these species in downstream analyses.

### 2.5. Microbe Abundance Correlation to PTC-Specific Oncogenic Pathways

Given the unique microbial differences we characterized across PTC cohorts and gender comparisons, we correlated clinically relevant fungal species to known oncogenic drivers of PTC. In particular, we focused on the BRAF, RET, P53, RAS, MAPK, and AKT pathways, all of which are implicated in PTC initiation and progression [41,42,43,44,45,46,47,48]. We used Gene Set Enrichment Analysis (GSEA) to correlate fungal microbial abundance to the following gene set signatures available on the Molecular Signature Database (https://www.gsea-msigdb.org/gsea/msigdb/index.jsp), accessed on 7 November 2022: (1) PID_PI3KCI_AKT_PATHWAY, (2) BIOCARTA_RAS_PATHWAY, (3) REACTOME_SIGNALING_BY_MODERATE_KINASE_ACTIVITY_BRAF_MUTANTS, (4) KEGG_P53_SIGNALING_PATHWAY, (5) REACTOME_RET_SIGNALING, and (6) REACTOME_ONCOGENIC_MAPK_SIGNALING. In order to characterize the association of the intratumor mycobiome with these oncogenic PTC pathways, we correlated clinically relevant fungal species with these pathways according to the cohorts in which they were significantly dysregulated. 

First, we correlated fungal microbial abundance to oncogenic PTC pathways in CPTC patients (Figure 5A). We found that *Metschnikowia santaceciliae*, *Pacynthium nigrum*, *Thanatephorus cucumeris*, and *Spriromyces aspiralis* were correlated with negative enrichment of PI3K/AKT pathway, while *Metschnikowia santaceciliae* and *Placynthium nigrum* were associated with negative enrichment of the RAS signaling pathway. Conversely, we observed that *Uncultured Galactomyces* was correlated with positive enrichment of BRAF kinase activity, indicating a potential oncogenic role of this species in CPTC. The majority of significantly enriched pathways were negatively enriched in CPTC patients, suggesting these microbes may play a tumor-suppressive role in CPTC.

Next, we associated fungal microbes with known oncogenic PTC pathways in TCPTC patients (Figure 5B). We found three fungal species, *Brevicellicium exile, Eremascus albus,* and *Zoophthora occidentalis*, associated with positive enrichment of p53 signaling, and two fungal species, *Metschniokowia santaceciliae* and *uncultured Glomus*, correlated with positive enrichment of BRAF kinase activity. *Brevicellicium exile*, *Metschniokowia santaceciliae*, and *uncultured Glomus* were also correlated with positive enrichment of RET, MAPK, and RAS signaling, indicating these microbes may play a significant role in TPTC initiation and progression through multiple oncogenic pathways. 

In patients with FVPTC, we found species categorized as *uncultured Glomus* were associated with positive enrichment of BRAF kinase activity and MAPK signaling, and *Rozella allomycis* was correlated with positive enrichment of only the BRAF kinase activity signature (Figure 5C). These findings suggest that *uncultured Glomus* may be pertinent to FVPTC oncogenesis, while significant enrichment via other microbes was not observed. Further studies are needed, however, to culture *Glomus* species and elucidate which specific microbes may be associated with the enrichment of these pathways.

Finally, we identified enriched oncogenic pathways in male and female patients according to fungal species abundance (Figure 5D). We did not observe any significant pathway enrichment in male patients. In female patients, however, *Coemansia reversa* and *Spiromyces aspiralis* were associated with positive enrichment of BRAF kinase activity, and *Coemansia reversa* exhibited a positive correlation with RET signaling. Moreover, *Spiromyces aspiralis* was also associated with positive enrichment of RET signaling.

In all, GSEA analysis showed unique correlations with oncogenic pathways implicated in PTC, indicating that these fungal microbes may play a unique role in occurrence and oncogenesis through different mechanisms in each PTC subtype and gender.

### 2.6. Differential Abundance according to BRAF 600VE Mutation Status

Given the unique oncogenic components associated with these microbes, we also conducted differential abundance analysis according to the incidence of the BRAF V600E mutation (Figure 6). The BRAF V600E mutation is the most common genetic alteration in PTC, initiating tumorigenic events through the mitogen-activated protein kinase (MAPK) signaling transduction pathway [49,50]. 

In our analysis, we found 18 fungal species overabundant in tumor tissue with no BRAF V600E present, including *Phaeoacremonium minimum UCRPA7* and *Saccharomyces cerevisiae YJM1615* (Figure 6A,B). Only six fungal species were overabundant in tumor tissue with the BRAFV600E mutation, including *Volvariella volvacea*, and *Metarhizium anisopliae*. Thus, these species appear to be overabundant, primarily in BRAF V600E-negative tissues. Further studies are needed to validate the presence of these microbes and the mechanisms by which they may confer oncogenic activity in vitro. 

## 3. Discussion

In this study, we characterized the intratumor mycobiome and archaeometry in PTC according to CPTC, TCPTC, and FVPTC subtypes and gender comparisons. Previously, we investigated the intratumor bacterial microbiome in PTC according to these cohorts [51]; our findings suggested that a unique microbial landscape exists in PTC according to these factors and is furthermore associated with distinct immune-associated, oncogenic, mutational, and methylation elements. 

To our knowledge, we are the first to associate the abundance of fungal species within the tumor microenvironment with important prognostic variables and oncogenic signatures. Recently, a pan-cancer study of the mycobiome in 35 cancers demonstrated the relevance of these fungal microbes for diagnostic purposes, developing a highly-accurate machine learning model from these microbial elements in thyroid and many other cancers [52]. While this study was one of the first to associate the fungal mycobiome with PTC using data from TCGA [52], specific characterization of the intratumor PTC mycobiome according to key diagnostic and prognostic factors, such as subtype and gender, was not pursued due to the scale of the study. Thus, our study provides important insights into how the dysregulation of the mycobiome may influence PTC oncogenesis through pathways distinct to patient subtype and gender. Further in vitro experimentation is needed, however, to validate these results and elucidate the specific metabolic mechanisms which may lead to these oncogenic effects.

Additionally, our study explores a novel aspect of the human microbiome in cancer. While most investigations of the microbiome in cancer focus on bacteria and fungi, archaeal species are rarely studied in human disease. Currently, most studies of archaeometry analyze species abundant in the gut, which consist primarily of methanogens [53,54]. A recent study, however, found archaeal species abundant in organs with high exposure to the environment, including the lung, nose and skin [55]. So far, studies of the archaeome in cancer have been limited to colorectal cancer [56,57]. Thus, our investigation of the archaeometry in lung adenocarcinoma (LUAD) and lung squamous cell carcinoma (LUSC) was the first to associate archaeal microbes with pertinent metabolic and oncogenic pathways in any non-colorectal cancer [58]. In this paper, we found that intratumor archaeal species were distinctly dysregulated between LUAD and LUSC samples and exhibited associations with unique cellular pathway regulation and clinical progression variables. Thus, we aimed to investigate a similar potential correlation between the intratumor PTC archaeometry and known oncogenic pathways in PTC, in addition to the association with prognostic variables. 

Overall, fungal species were disproportionately abundant in our samples compared to archaeal species. Interestingly, our findings suggest that higher dysregulation of fungal microbes exists in PTC tissue compared to the bacterial dysregulation we previously identified [51]. A large proportion of fungal microbes identified in our study were similarly overabundant in CPTC, FVPTC, and TCPTC, indicating that these microbes may be more dysregulated between cancer and normal samples than between subtypes of PTC. Additionally, the vast majority of these dysregulated microbes were overabundant in PTC tumor tissue compared to adjacent normals, indicating that intratumor PTC mycobiome may largely play a cancer-promoting role. Despite significant commonalities in fungal microbe composition across PTC subtypes, however, CPTC, FVPTC, and TCPTC still exhibited noticeable differences in dysregulated species. TCPTC, for example, exhibited the most amount of dysregulation of the three subtypes, followed by FVPTC and CPTC: with particularly disproportionate upregulation of dysregulated microbes. Unlike the FVPTC and TCPTC variants, however, microbes uniquely dysregulated in CPTC tumor tissue were primarily underabundant compared to normal samples. Thus, uniquely dysregulated microbes of the intratumor microbiome in CPTC may serve a tumor-suppressive role, while fungal species unique to TCPTC and FVPTC may be implicated more significantly in oncogenic functionalities. Our analysis of fungal microbe abundance with oncogenic pathway enrichment suggested similar results, with negative enrichment of these pathways associated with fungal species only observed in CPTC patients. Future in vitro studies are needed to confirm these microbial differences in PTC subtypes and further characterize their role in oncogenic events. 

While our previous study of the intratumor bacterial PTC microbiome found almost comparable differences in abundance between male and female tissue samples [51], we found significant differences in the fungal mycobiome according to gender. Although a higher proportion of dysregulate fungal microbes was common to all study types of PTC, most dysregulated fungal microbes in gender comparisons were distinct to males and females. Female tumor samples exhibited 80 uniquely overabundant microbes compared to normal samples, including *Pseudosperma obsoletum*, *Yamadazyma triangularis,* and *Candida albicans*. We found that only four species were uniquely overabundant in male tumor samples: *Rhizomucor miehei*, *Didymella macropodii*, *Geranomyces variabilis*, *Vittaforma cornaea ATCC 50505.* As such, our findings suggest that differences in the intratumor PTC mycobiome are likely more noticeable in gender comparisons than in subtypes. 

Few archaeal species were found to be dysregulated in our analysis. However, these species were unique to CPTC, FVPTC, and TCPTC tissue. Additionally, archaea were only found to be dysregulated in males, including *Halovivax ruber XH*-70 and *Natrialba magadii ATCC 43099.* Thus, few differences in archaeal abundance were observed according to these cohorts. However, further studies and culturing of archaeal species may reveal even more species abundant in thyroid tissue. The lack of archaeal species in the thyroid tissue may be due to its relatively lower exposure to the environment compared to other organs, such as the mouth, lungs, and nose. 

Additionally, we correlated these dysregulated fungal microbes to clinical progression and vital status variables in all PTC patients. *Brevicellium exile* was overabundant in patients who were deceased at the last follow-up. The majority of our clinical variable analysis indicated that many of these microbes are pertinent to the pathologic stage, particularly the pathologic M stage. The abundance of *Chaetomium globosum CBS 148.51,* for example, was correlated with a higher overall pathologic stage. The abundance of 18 microbial species, including *Candida albicans* and *Eremascus albus,* were correlated with higher pathologic M stage, suggesting that these fungal microbes may be additionally pertinent in PTC tumor metastasis. 

In order to further characterize the role of these microbes in PTC oncogenesis, we associated dysregulated fungal microbes with known pathways implicated in PTC development, including the BRAF, P53, RET/PTC, MAPK, KAT, and RAS signature pathways. Consistent with our findings of underabundance in CPTC tumor tissue compared to normals, we found that the majority of these pathways were negatively enriched in CPTC, primarily PI3K/AKT, which induces tumor growth and energy storage of cancer cells [59,60]. Unlike in TCPTC and FVPTC, microbes dysregulated in CPTC appeared to be associated with the inhibition of cancer-promoting pathways. Enrichment of the BRAF and P53 signaling pathways, however, was associated with multiple fungal species in TCPTC. Fewer pathways were enriched by microbes in FVPTC; however, BRAF and MAPK signaling were positively enriched by *uncultured Glomus*. Interestingly, no pathways were significantly enriched by dysregulated microbes in male patients. Similar to TCPTC and FVPTC, the BRAF pathway was enriched by multiple fungal microbes in females. Interestingly, RET signaling was not associated with enrichment via microbe abundance nearly as much as other studied pathways, including BRAF, P53, MAPK, and AKT. The RET/PTC arrangement is a hallmark of PTC development and is the most common genetic alteration in PTC [61]. Thus, future studies should validate if the microbiome is implicated in PTC oncogenesis through pathways other than RET signaling. Additionally, further in vitro experimentation is needed to understand the exact metabolic interactions of intratumor microbes with these signaling pathways. 

We also found several microbes were dysregulated in tumor samples with the BRAF V600E mutation. The BRAF V600E mutation is the most commonly mutated oncogene in PTC, which initiates tumorigenesis via activation of the MAPK signaling pathway [50]. Interestingly, we found that the majority of dysregulated microbes, according to BRAF V600E mutation status, were overabundant in BRAF V600E-negative tissue. Thus, it is plausible that this mutation may also dysregulate fungal microbes within the tumor microenvironment; however, further studies are needed to elucidate this mechanistic role in PTC. 

In all, our findings suggest that the intratumor mycobiome and archaeometry in PTC differ greatly according to subtype and gender. To the best of our knowledge, we are the first to associate these microbial elements in PTC with pertinent prognostic variables. Additionally, the abundance of fungal species exhibited correlations to higher pathologic staging, particularly metastasis, and unique correlations to known oncogenic PTC pathways in CPTC, FVPTC, TCPTC, and females. Due to the correlative nature of our study, in vitro analysis must be conducted to confirm the role of these microbes in PTC incidence and progression. Although CPTC, FVPTC, and TCPTC were the primary tumors available from TCGA, future studies should also characterize how these microbes may uniquely contribute to oncogenesis in other forms of thyroid cancer [62,63,64]. 

## 4. Materials and Methods

### 4.1. Data Acquisition

Raw whole-transcriptome RNA-sequencing data were downloaded from TCGA-THCA project (https://portal.gdc.cancer.gov/projects/TCGA-THCA), accessed on 15 January 2022, for 453 thyroid carcinoma primary tumor samples and 54 adjacent solid tissue normals. Clinical data for the patients investigated in this study were obtained from the Broad Genome Data Analysis Center (GDAC) Firehose database (https://gdac.broadinstitute.org/), accessed on 3 January 2023. All data analyzed in this study were accessed and analyzed during the period January 2022–December 2022.

### 4.2. Extraction and Normalization of Fungal and Archaeal Read Counts

Fungal and Archaeal read counts were extracted from raw RNA-sequencing data using the PathoScope 2.0 alignment tool and the NCBI nucleotide database. Microbial reads were successfully extracted from all 507 RNA-sequencing files. In order to normalize data and reduce variance across samples, we conducted Aitchison’s log transformation on all extracted read counts.

### 4.3. Evaluation of Contamination

In order to account for potential contaminants introduced through sequencing or sampling methods conducted on our samples, we conducted contamination correction by sequencing date, plate, and microbial abundance. First, we conducted contamination correction by sequencing date by creating scatter plots of the sequencing date compared to microbial abundance for each microbe. Microbes with a scatter plot exhibiting one functional abundance cluster on a certain date were considered potential contaminants. Second, we conducted contamination correction by sequencing plate by plotting boxplots of microbial abundance according to each respective sequencing plate. Microbes that were disproportionately abundant in two or fewer sequencing plates were identified as potential contaminants. Finally, contamination correction by total microbe abundance was conducted by creating scatter plots of total microbial abundance (global abundance of microbes in each patient) compared to abundance of each individual species. We identified potential contaminants in plots with a slope of zero (margin of error ± 0.1). Visual identification of contaminants was verified by at least two authors for each contamination correction method. A complete list of identified contaminants, including those not visualized in Figure 2, is included in Appendix A.

### 4.4. Differential Abundance between PTC, Gender, and Mutation Cohorts

Differential abundance analysis was conducted on the following patient cohorts: (1) primary tumor samples and adjacent normal samples, (2) CPTC samples and adjacent normal samples, (3) FVPTC samples and adjacent normal samples, (4) TCPTC samples and adjacent normal samples, (5) male cancer samples and male adjacent normal samples, and (6) female cancer samples and female adjacent normal samples. Differential abundance analysis was conducted for each of these cohorts for fungal and archaeal data, using the Kruskal–Wallis test in the edge-R library. Statistically significant results were defined with a *p*-value < 0.05. We then identified similar differentially-abundant microbes across PTC subtypes and gender comparisons according to the direction of overabundance. Significantly dysregulated microbes were used in further analyses. A complete list of significantly dysregulated microbes, including those not visualized in Figure 3, is included in Appendix A.

Similarly, we also conducted differential abundance analysis on BRAF V600E positive cancer samples and BRAF V600E negative cancer samples. Clinical information for PTC subtype, gender, and mutation status was extracted from the GDAC Firehose clinical data file.

### 4.5. Association of Microbial Abundance to Clinical Variable

We assessed significantly dysregulated microbes to clinical variables using the Kruskal–Wallis test (*p*-value < 0.05). In order to correct for multiple hypothesis testing, we adjusted our *p*-values using the Bonferroni method. We examined six main clinical variables pertinent to PTC prognosis and clinical course: follow-up vital status, perineural invasion, pathologic stage, and cancer T, N, and M staging. We performed clinical variable analysis on all patients with PTC (n = 453). 

### 4.6. Correlation of Microbial Abundance to Oncogenic PTC Signature Pathways

We conducted Gene Set Enrichment Analysis (GSEA) to correlate microbial abundance to known pathway signatures implicated in PTC oncogenesis. Three input files were prepared for GSEA analysis: the expression file, the phenotype file, and the gene set file. The expression file consisted of gene expression data, and the phenotype file contained microbial abundance features for each sample. The geneset file was created with the following signature pathways defined by the Broad Institute: (1) PID_PI3KCI_AKT_PATHWAY, (2) BIOCARTA_RAS_PATHWAY, (3) REACTOME_SIGNALING_BY_MODERATE_KINASE_ACTIVITY_BRAF_MUTANTS, (4) KEGG_P53_SIGNALING_PATHWAY, (5) REACTOME_RET_SIGNALING, (6) REACTOME_ONCOGENIC_MAPK_SIGNALING. The specific classification file is included in Appendix A. Only statistically significantly enriched signatures were further analyzed (*p*-value < 0.05) and false discovery rate (FDR) < 0.25). 

## 5. Conclusions

In conclusion, our study provides novel insights into the potential importance of fungal and archaeal species to oncogenesis within the PTC microenvironment. Additionally, we characterized important similarities and differences in the intratumor PTC mycobiome and archaeometry according to PTC subtype (classical, follicular variant, and tall cell) and gender. Overall, the majority of dysregulated species in PTC samples were overabundant in tissue. A total of 63 fungal species were commonly overabundant in CPTC, FVPTC, and TCPTC, including *Botrytis cinerea,* while 33 fungal species were uniquely overabundant in TCPTC, and 28 in FVPTC, whereas 16 fungal species were uniquely underabundant in CPTC. This collection of microbes includes *Phialophora verrucosa, Boletinellus merulioides*, and *Bipolaris sorokiniana*, respectively. We found that the fungal and archaeal landscapes, however, were more distinct across gender comparisons. Amongst 80 total microbes, *Pseudosperma obsoletum* and *Candida albicans* were uniquely overabundant in female PTC tumor tissue. Archaeal species were uniquely dysregulated according to PTC subtypes and gender. *Halovivax ruber XH*-70 and *Natrialba magadii ATCC 43099* were uniquely underabundant in male PTC tumor tissue compared to male-controlled adjacent normal tissue. In clinical variable analyses, several fungal microbes were associated with higher pathologic staging, including *Candida albicans* and *Eremascus albus*. CPTC was characterized primarily by negative enrichment of oncogenic PTC pathways, including the PI3K/AKT signaling pathway. Conversely, the P53 and BRAF mutant pathways were positively enriched by several microbes in TCPTC, FVPTC, and females, including *uncultured Glomus, Eremascus albus,* and *Metschnikowia santaceciliae*. Finally, we found that *Volvariella* volvacea was overabundant in BRAF V600E-positive tumors, while *Phaeoacremonium minimum* UCRPA7 was overabundant in BRAF V600E-negative tumors. Future in vitro experiments are needed to validate these microbial differences and associations to clinical variables and pertinent cancer pathways. 

## Figures and Tables

**Figure 1 ijms-24-03148-f001:**
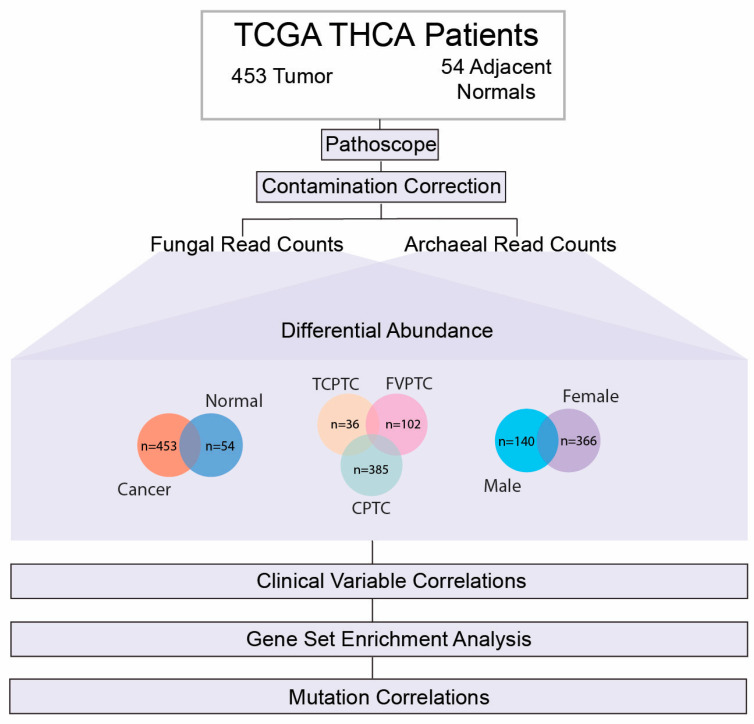
Schematic of analysis pipeline used for this study. Raw RNA-sequencing data were downloaded from The Cancer Genome Atlas (TCGA) for 453 PTC patients and 54 adjacent normal samples. The PathoScope 2.0 software was used to align RNA-sequencing files to NCBI nucleotide database’s reference genomes and extract fungal and archaeal read counts. Following sequencing alignment, multiple contamination correction methods were used to identify and remove potential contaminants from downstream analyses. In order to characterize the fungal and archaeal microbes pertinent to PTC, we first compared differentially abundant species in following cohorts: (1) 453 PTC samples and 54 adjacent normal samples; (2) 385 CPTC, 102 FVPTC, and 36 TCPTC samples; (4) 140 male cancer samples and 366 female cancer samples, utilizing Kruskal–Wallis testing (*p*-value < 0.05). Following differential abundance analysis, significantly dysregulated microbes were correlated to clinical variables pertinent to PTC prognosis and outcomes. Dysregulated microbes were also correlated to known signature pathways specific to PTC oncogenesis through Gene Set Enrichment Analysis (GSEA). Finally, we correlated abundance of these microbes according to status of the BRAF V600E mutation in our patient cohort.

**Figure 2 ijms-24-03148-f002:**
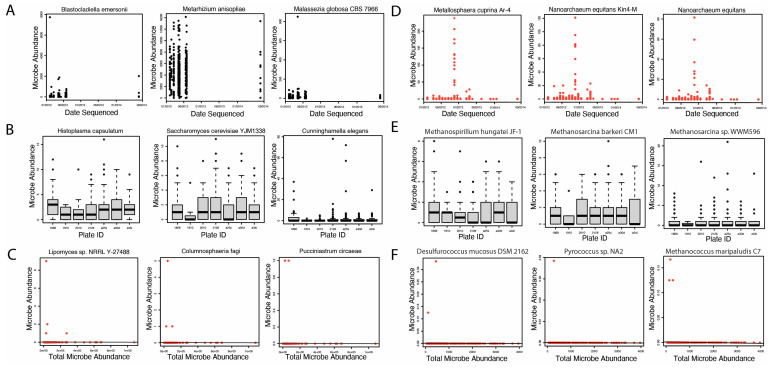
Selected plots from contamination correction analysis. Plots with red datapoints represent microbes that were identified as potential contaminants. A full list of contaminants for each contamination correction method is available in Appendix A. (**A**) Selected scatter plots for fungal species assessed for contamination correction by sequencing date. The date of sampling sequencing is represented on the x-axis, and microbe abundance is represented on the y-axis. Potential contaminants were defined as microbial plots with an abundance cluster around one specific date. No potential fungal contaminants were identified using this contamination correction method. (**B**) Selected boxplots for fungal species assessed for contamination correction by sequencing plate. Sequencing plates are represented on the x-axis, and microbe abundance is represented on the y-axis. Potential contaminants were defined as microbial plots in which the abundance for a particular sequencing plate was unusually high compared to others. No potential fungal contaminants were identified using this contamination correction method. (**C**) Selected scatter plots for fungal species assessed for contamination correction by total microbial abundance. Total microbial fungal microbe abundance is represented on the x-axis, while microbial abundance for a specific microbe is represented on the y-axis, for each patient. Potential contaminants were defined as microbial plots in which the linear regression analysis exhibited a slope of zero (margin of error ± 0.1). A total of 1492 fungal species were identified as potential contaminants when compared to total microbial abundance. Selected plots of three potential contaminant species are shown here. (**D**) Selected scatter plots for archaeal species assessed for contamination correction by sequencing date. A total of 6 archaeal species were identified as potential contaminants according to sequencing date. (**E**) Selected boxplots for archaeal species assessed for contamination correction by sequencing plate. No potential archaeal contaminants were identified using this contamination correction method. (**F**) Selected scatter plots for archaeal species assessed for contamination correction by total microbial abundance. A total of 286 archaeal species were identified as potential contaminants when compared to total microbial abundance. Selected plots of three potential contaminant species are shown here.

**Figure 3 ijms-24-03148-f003:**
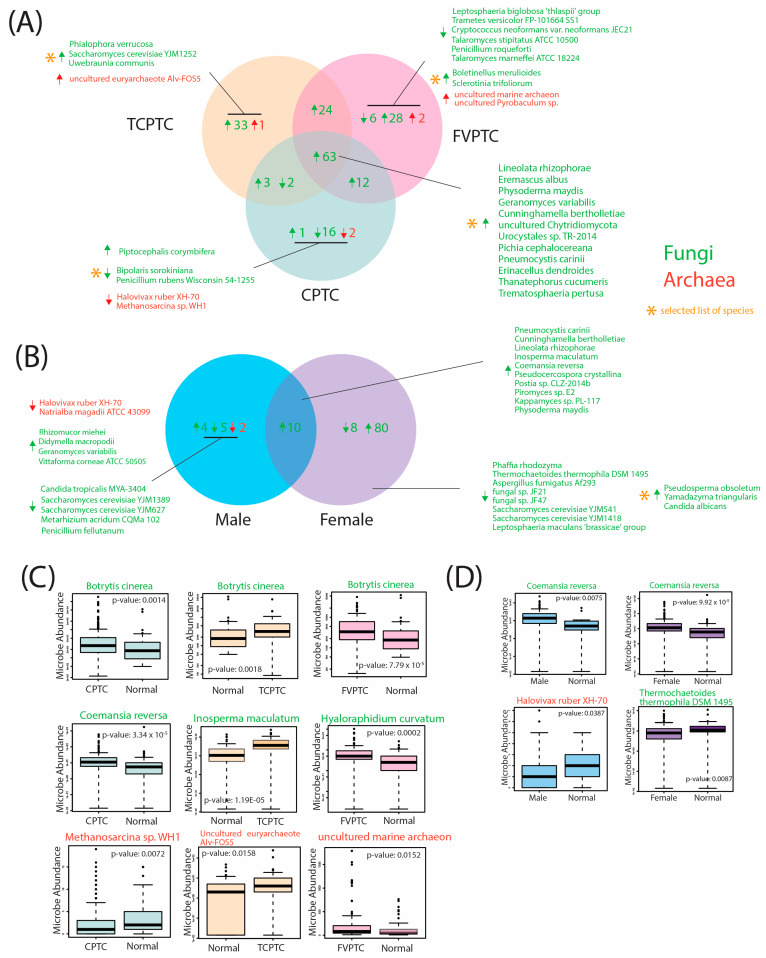
Characterization of the mycobiome and archaeome across PTC subtype and gender comparisons suggest unique dysregulation between these cohorts. Differential abundance analysis was performed in order to compare abundance of fungal and archaeal species in CPTC, TCPTC, FVPTC, male, and female tumor tissue samples to adjacent normal and gender-controlled adjacent normal samples. Significantly differentially abundant microbes were defined by a *p*-value of <0.05 and a log-fold change (FC) > 1. Fungal and archaeal species are displayed in green and red fonts, respectively. Arrows were used to visualize the direction of abundance: upward-facing arrows indicate overabundance of microbial species in the specified cohort(s), downward-facing arrows indicate underabundance of microbial species in the specified cohort(s). Asterisks indicate that only certain microbes in the comparison group were visualized in the figure. A complete list of significantly dysregulated microbes were included in Appendix A. (**A**) Venn diagram displaying selected commonly significantly dysregulated microbes and the direction of abundance between CPTC, TCPTC, and FVPTC tumor tissue samples compared to adjacent normal tissue. (**B**) Venn diagram displaying selected commonly significantly dysregulated microbes and the direction of abundance between male and female PTC tumor tissue samples compared to adjacent normal tissue. (**C**) Selected boxplots of significantly dysregulated microbes from PTC subtype analysis. *p*-values obtained from the Kruskal–Wallis test are indicated on each microbe plot. (**D**) Selected boxplots of significantly dysregulated microbes from gender comparison analysis. *p*-values obtained from the Kruskal–Wallis test are indicated on each microbe plot.

**Figure 4 ijms-24-03148-f004:**
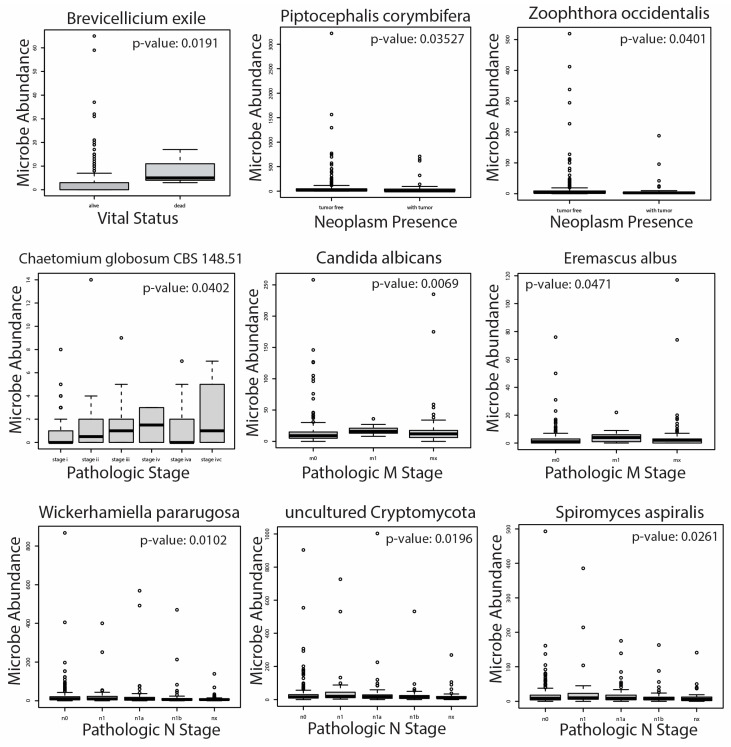
Selected boxplots displaying the correlation of significantly dysregulated microbes in PTC patients (n = 453) with clinical variables pertinent to cancer prognosis and outcomes. We performed clinical variable analysis using the Kruskal–Wallis test (*p*-value < 0.05) on fungal and archaeal microbes and following clinical variables: follow-up vital status, perineural invasion, pathologic stage, and cancer T, N, and M staging. In total, we found 26 microbes were significantly correlated with these clinical variables (Appendix A), all of which were fungal species.

**Figure 5 ijms-24-03148-f005:**
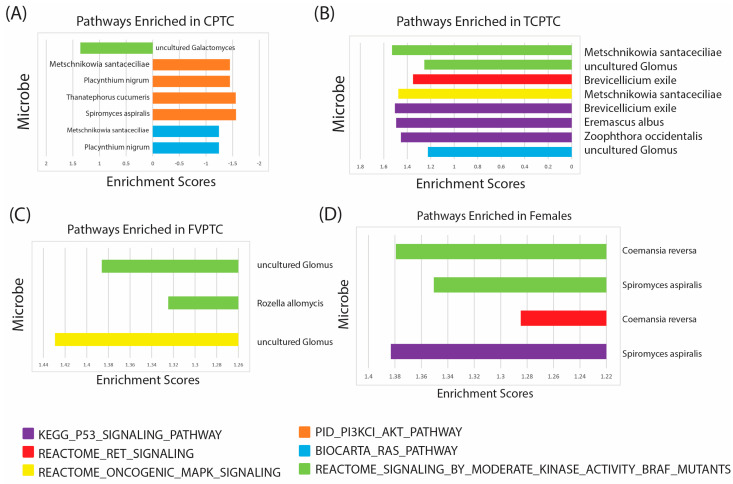
Enriched oncogenic signatures associated with dysregulated microbes in PTC and gender cohorts. Gene set enrichment analysis (GSEA) was conducted to identify unique oncogenic PTC signatures associated with dysregulated microbes pertinent to clinical variables (*p* < 0.05 and false discovery rates (FDRs) < 0.25). (**A**) Bar plots visualizing enriched pathways associated with fungal microbe species in CPTC patients. A positive enrichment score indicates pathway enrichment in CPTC patients. (**B**) Bar plots visualizing enriched pathways associated with fungal microbe species in TCPTC patients (**C**) Bar plots visualizing enriched pathways associated with fungal microbe species in FVPTC patients (**D**) Bar plots visualizing enriched pathways associated with fungal microbe species in female patients. No significantly enriched pathways were observed in male patients.

**Figure 6 ijms-24-03148-f006:**
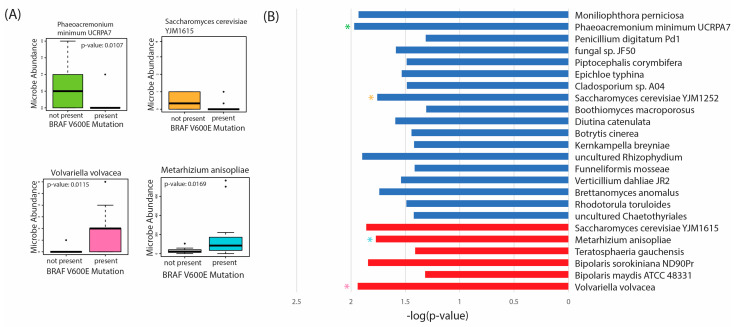
Differential abundance analysis of fungal species according to BRAF V600E mutation status in PTC patients. Significantly differentially abundant microbes were defined by a *p*-value of < 0.05 and a log-fold change (FC) > 1. (**A**) Selected boxplots of differentially abundant microbes in patients with the BRAF V600E mutation vs patients with no mutation. (**B**) −log(*p*-value) of sinificantly dysregulated microbes according BRAF V600E mutation status. A total of 18 fungal species, represented in blue were overabundant in BRAF V600-negative patients, while 6 fungal species, represented in red, were overabundant in BRAF V600-positive patients. Asterisks represent corresponding boxplots, according to color, represented in Part A.

## Data Availability

Publicly available datasets were analyzed in this study. Data can be found here: https://portal.gdc.cancer.gov/projects/TCGA-THCA, accessed on 3 January 2023.

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
