# Peer review of "Transcriptomic Analysis Reveals Dysregulation of the Mycobiome and Archaeome and Distinct Oncogenic Characteristics according to Subtype and Gender in Papillary Thyroid Carcinoma"

_ijms, 2023, doi:10.3390/ijms24043148_

Round 1

Reviewer 1 Report

Dear authors, 

 The manuscript you proposed for publication is up-to-date and well-written. Excellent statistics analysis! I suggest the authors to include the following paper, which contains very interesting information: https://doi.org/10.3390/life12091314.

Author Response

Please see the file attached.

Reviewer 2 Report

 Dear Authors,

The title of the manuscript misleads to in vivo or in vitro study. It is better to explain in the title. 

The period of the study did not mention. The data is close to the publication of Gnanasekar A et al. The intratumor microbiome predicts prognosis across gender and subtypes in papillary thyroid carcinoma. Comput Struct Biotechnol J. 2021).

The study identifies the intratumor fungal and archaeal microbes according to PTC subtype and gender comparisons. Then the authors identify them as dysregulated mycobiome when the study basis on software analysis. The authors should identify specific nature regulations of mycobiome first before identifying it as dysregulated. 

Round 2

Reviewer 2 Report

The authors improve the quality of manuscript.